# The GPI-Anchored Protein Thy-1/CD90 Promotes Wound Healing upon Injury to the Skin by Enhancing Skin Perfusion

**DOI:** 10.3390/ijms232012539

**Published:** 2022-10-19

**Authors:** Leonardo A. Pérez, José León, Juan López, Daniela Rojas, Montserrat Reyes, Pamela Contreras, Andrew F. G. Quest, Carlos Escudero, Lisette Leyton

**Affiliations:** 1Laboratory of Cellular Communication, Center for Studies on Exercise, Metabolism and Cancer (CEMC), Faculty of Medicine, University of Chile, Santiago 8380453, Chile; 2Advanced Center for Chronic Diseases (ACCDiS), Faculty of Medicine, Universidad de Chile, Santiago 8380453, Chile; 3Vascular Physiology Laboratory, Department of Basic Sciences, Faculty of Sciences, Universidad del Bío-Bío, Chillán 3800708, Chile; 4Escuela de Enfermería, Facultad de Salud, Universidad Santo Tomás, Los Ángeles 3460000, Chile; 5Departamento de Patología Clínica, Facultad de Ciencias Veterinarias, Universidad de Concepción, Chillán 4030000, Chile; 6Department of Pathology and Oral Medicine, Facultad de Odontología, Universidad de Chile, Santiago 8380544, Chile; 7Group of Research and Innovation in Vascular Health (GRIVAS Health), Chillán 3800708, Chile

**Keywords:** wound healing, Thy-1 (CD90), angiogenesis, skin injuries, blood perfusion, Fourier analysis

## Abstract

Wound healing is a highly regulated multi-step process that involves a plethora of signals. Blood perfusion is crucial in wound healing and abnormalities in the formation of new blood vessels define the outcome of the wound healing process. Thy-1 has been implicated in angiogenesis and silencing of the Thy-1 gene retards the wound healing process. However, the role of Thy-1 in blood perfusion during wound closure remains unclear. We proposed that Thy-1 regulates vascular perfusion, affecting the healing rate in mouse skin. We analyzed the time of recovery, blood perfusion using Laser Speckle Contrast Imaging, and tissue morphology from images acquired with a Nanozoomer tissue scanner. The latter was assessed in a tissue sample taken with a biopsy punch on several days during the wound healing process. Results obtained with the Thy-1 knockout (*Thy-1*^−/−^) mice were compared with control mice. *Thy-1***^−^**^/−^ mice showed at day seven, a delayed re-epithelialization, increased micro- to macro-circulation ratio, and lower blood perfusion in the wound area. In addition, skin morphology displayed a flatter epidermis, fewer ridges, and almost no stratum granulosum or corneum, while the dermis was thicker, showing more fibroblasts and fewer lymphocytes. Our results suggest a critical role for Thy-1 in wound healing, particularly in vascular dynamics.

## 1. Introduction

Lack of healing is a significant problem in skin wounds because it contributes to chronic ulcers, inflammation, and persistent infections, among other pathological manifestations [1]. Wound healing involves multiple processes, including extracellular matrix remodeling, synthesis of pro-inflammatory mediators, and the formation of new vessels from preexisting vessels (or angiogenesis) [2,3,4].

The formation of new blood vessels promotes proper tissue perfusion and wound healing [4]. Therefore, impaired vascular function, including alterations in the angiogenesis process or blood perfusion of the tissue, compromises dermal wound healing outcomes [4,5]. Analysis of dermal blood perfusion represents a challenge due to the involvement of small blood vessels (i.e., microcirculation) and high dynamism.

Several invasive and non-invasive approaches have been described in the literature to analyze dermal blood perfusion, including Laser Doppler blood flowmetry (LDF) [6]. Laser speckle contrast imaging (LSCI) enables non-contact, real-time, and non-invasive monitoring of changes in the blood perfusion of the skin [7,8]. This technique involves imaging time-integrated speckle patterns generated by low-power laser irradiation that is captured by a high spatiotemporal resolution charge-coupled device (CCD) camera. In addition, depending on the experimental setting, the LSCI technique generates a large amount of data that requires sophisticated data analysis. Fourier transform (FT) can be used as the signal processing method of skin−blood perfusion to decompose perfusion signals in the spectral domain, which may reveal various physiological rhythms associated with blood flow control mechanisms [9]. For instance, wavelet analysis of LDF signals from the human forearm or feet has revealed five characteristic frequency bands, corresponding to heartbeat dynamics; rhythmicity of breath; rhythmic activity of vessels; and metabolic activity [9,10]. Analysis of the relative contribution of these wavelets may shed light on the underlying vascular function and reactivity.

Thy-1(CD90) is a glycosyl phosphatidyl inositol (GPI)-anchored protein that resides in lipid rafts. It is essential in cell migration and is strongly upregulated in endothelial cells during pro-inflammatory cytokine-induced angiogenesis [11,12,13,14]. These observations have indicated that Thy-1 might be relevant to the process of wound healing following lesions to the skin. Indeed, neutrophils cross the endothelium during the inflammatory phase and target the injured area in a Thy-1-dependent manner [14]. Furthermore, the adhesion of neutrophils to activated endothelial cells with high Thy-1 expression promotes the binding of these cells [15] through Thy-1/Mac-1(CD11b-CD18; integrin αMβ2) interaction [16].

In recent years, increased emphasis has been granted to studying the role of Thy-1 and its downstream signaling pathways in wound healing [17]. Specifically, Lee and co-workers [18] investigated these events using *Thy-1* knockdown in a mouse model of skin wound healing and showed that wound repair is retarded when Thy-1 levels are decreased. So far, only a few studies have suggested a clear association between Thy-1 expression and angiogenesis [18,19]; these have reported that Thy-1 expression is crucial in the process of new vessel formation. We recently reviewed the role of Thy-1 and its receptors—integrins and Syndecan-4—in wound healing [17]. From this analysis of the literature, it became clear that the function of Thy-1 in the angiogenic process remains poorly understood.

This study explored the role of Thy-1 in a pre-clinical wound healing model using *Thy-1* knockout (*Thy-1*^−/−^) mice and evaluated the timeline of wound closure, microvessel number in the wounded area, blood perfusion, and tissue remodeling. We also analyzed the relative contribution of three wavelet components of blood perfusion (i.e., metabolic, neurogenic, and myogenic wavelets) during the wound healing process. We provide evidence showing that Thy-1 regulates vascular perfusion, thereby affecting the healing rate in mouse skin.

## 2. Results

### 2.1. Lack of Thy-1 Expression Delays Wound Healing

Previous studies have indicated that the lack of Thy-1 in a localized area of a knockdown mouse model delays the wound healing process, suggesting a role for Thy-1 in wound repair [18]. We confirmed these findings with a proof-of-concept experiment using the *Thy-1* knockout (*Thy-1*^−/−^) mouse model. First, we challenged wild-type (WT) and *Thy-1*^−/−^ mice with a wound biopsy punch of 2 mm in the head skin between the ears; the day of the biopsy punch was called T0. The wounded area was measured 4 and 7 days after the T0 (Figure 1A). We found that the wound was almost 100% closed in WT mice on day 7, while *Thy-1*^−/−^ mice showed a significant delay in the wound healing process compared with the WT group on days 4 and 7 (Figure 1B,C).

### 2.2. Skin Morphology during Wound Healing Is Altered in Mice Lacking Thy-1

Skin morphology exhibits layers that protect our body from foreign threats, where the epidermis, dermis, and hypodermis form part of the human body’s largest organ [20]. Regarding tissue morphology, even though the wound was nearly closed after seven days in the WT group, the skin tissue was still undergoing remodeling (compare with normal skin, Figure 1B, see arrows). Furthermore, the tissue of the wound area in both mouse groups was analyzed by extracting the skin region with a bigger biopsy punch (4 mm) on days seven and fourteen (Figure 2A). The tissue was stained with hematoxylin-eosin to perform a morphological analysis (Figure 2B,C). Additionally, the thicknesses of the dermis and epidermis layers were measured in both groups (Figure 2D–G).

Histological observation of the injured and uninjured tissue samples, sections were stained with hematoxylin-eosin. The healthy mouse skin was organized into the epidermis with epithelial cells or keratinocytes in various stages of differentiation, the dermis, composed of extracellular matrix, fibroblasts, immune cells, and vascular elements, a variety of hair follicles, sebaceous glands, and sweat glands (Figure 2B). After an injury, the skin of the *Thy-1*^−/−^ mice exhibited a thinner epidermis than that of the WT mice. For *Thy-1*^−/−^ mice, seven days after wounding, the damaged area was constituted by a flat, multilayered epithelium and the cells in the stratum spinosum were arranged in a disorderly manner (Figure 2C). Moreover, compared to WT mice, not all the epidermal layers were present. Particularly, the granular layer and the stratum corneum were absent, and the basement membrane was discontinuous (Figure 2C). In addition, elements that are characteristic of the inflammatory phase were observed, with an increased number of fibroblast-like cells and fewer lymphocytes in the *Thy-1*^−/−^ model, likely indicating a prolonged inflammatory and proliferative phase than in WT mice. Extravascular red blood cells were also visualized (Figure 2C). The proliferative phase of wound repair was prolongated in *Thy-1*^−/−^ than in WT mice. After 14 days of wound healing, the size of the keratinized flat multilayered epithelium had remodeled to its original size in tissue sections from both animal models. In addition, we observed the first formations of hair follicles. No major histological differences were distinguished between the two groups at this stage of repair.

Quantitative analysis indicated a pronounced difference in re-epithelialization speed, and that wound closure was faster in WT than in *Thy-1*^−/−^ mice (Figure 1C). Quantification of the dermal and epidermal layers indicated that 7 days after injury, the epidermal layer exhibited similar thickness (around 340 μm) in both WT and *Thy-1*^−/−^ mice (Figure 2D). At the same time point, the dermis of *Thy-1*^−/−^ mice was, on average, slightly thicker than that of the WT animals. However, the differences were not statistically significant (Figure 2E). On the other hand, after fourteen days, the epidermis was noticeably thicker in WT, compared to *Thy-1*^−/−^ mice (Figure 2F); however, again, given the dispersion of the data in the WT model, the difference was not statistically significant. Finally, the dermal layer showed a significantly increased thickness in *Thy-1*^−/−^ mice, compared to the WT mice, fourteen days after injury (Figure 2G). Thus, our data indicated that the absence of Thy-1 reduces the speed of normal wound closure and changes the remodeling of the wounded tissue.

### 2.3. Micro and Macrovascular Vessels Are Regulated by Thy-1

Vascular irrigation is essential to restore skin function after tissue damage. Among blood vessels, we can differentiate macro and microvessels, which are defined depending on the size of the lumen. To determine the proportion and size of micro and macrovessels in WT and *Thy-1*^−/−^ mice, we analyzed histological samples 7- and 14-days post-injury (Figure 3). An increased percentage of microvessels was observed seven days after injury in *Thy-1*^−/−^ mice, compared with WT mice (Figure 3A).

The mean area of the macrovessel (Figure 3B) and microvessel (Figure 3C) lumen was similar in WT versus *Thy-1*^−/−^ mice. We also found a similar ratio (Figure 3D) and size of lumen in macro (Figure 3E) and micro (Figure 3F) vessels at day fourteen for both groups. Therefore, more microvessels are observed in the wounded area of *Thy-1*^−/−^ mice, compared with WT mice seven days post-injury.

### 2.4. Absence of Thy-1 Decreases Blood Perfusion in Wound Healing

Blood perfusion after an injury is crucial for successful wound healing and, while the blood clot is necessary to stop the bleeding, the bloodstream delivers nutrients, immune cells, and oxygen to the damaged area. Since *Thy-1*^−/−^ mice showed an increased number of microvessels in the wounded area on day seven after injury and considering that Thy-1 has been previously implicated in angiogenesis [19], we next wondered whether those blood vessels were functional. To this end, we performed an LSCI analysis in the wound area and the surrounding tissue in WT and *Thy-1*^−/−^ mice. Representative images of blood perfusion data from basal (prior to injury) and time zero (T0: immediate post-injury) are shown for both groups (Figure 4A). The wounded area was observed as an intense red circle, which, according to the color code bar, represents an area with high blood perfusion (Figure 4A). Perfusion quantification was performed prior (basal) and post-injury (T0) in the wounded area (inside the 2 mm diameter area) and the peripheral area (zone surrounding the wounded area).

Basally, *Thy-1*^−/−^ mice showed increased perfusion compared to WT mice when analyzing head skin areas, both directly where the punch had been performed (Figure 4B) and in the peripheral area (Figure 4C). However, at T0, *Thy-1*^−/−^ mice exhibited similar blood perfusion in the wound area (Figure 4D), and significantly higher perfusion in the peripheral area (Figure 4E) compared to WT mice.

Because differences in the basal levels of blood perfusion were detected between both groups (Figure 4B,C), we decided to normalize the data to the basal level in each animal per group to compare the values obtained on the indicated days after injury. Increased perfusion levels were observed in the wounded area on day one after injury in both groups of mice. However, the increase in WT mice was higher than in *Thy-1*^−/−^ mice (^##^
*p* < 0.01). This elevated perfusion was dynamic in WT mice since it remained significantly higher until day 10, compared to its respective basal condition, despite showing a continued decline until day 14. Notably, the increase in perfusion was significantly higher in the WT group, compared with the *Thy-1*^−/−^ group on days 1, 7, 10 and 14 after injury (Figure 5A). However, perfusion in the wounded area declined more in *Thy-1*^−/−^ than in WT mice at day 14 (Figure 5A). In contrast, blood perfusion in the peripheral area showed similar, unaltered behavior in both groups (Figure 5B), except on day 14 where perfusion was significantly lower in both models, compared to their respective basal conditions (Figure 5B).

The perfusion (wound and peripheral) signal data were transformed into the wavelet components using Fourier transformation to determine the participation of the metabolic (Figure 5C,F), myogenic (Figure 5D,G), and neurogenic components (Figure 5E,H) of blood perfusion in both WT and *Thy-1*^−/−^ mice at basal, T0, day four, day seven, and day fourteen after injury. A significant increase in the myogenic and neurogenic components of the peripheral perfusion signal was found in *Thy-1*^−/−^ mice immediately after skin injury (T0), compared with WT mice. Significant differences were also found in the metabolic and neurogenic components in both wounded and peripheral areas on day seven after injury, in which the *Thy-1*^−/−^ group exhibited higher participation of these components than WT mice. Furthermore, we also observed significant differences on days four and fourteen in the *Thy-1*^−/−^ group for the myogenic and neurogenic components, compared to the WT mice in the peripheral area. Together, these data indicate that Thy-1 is important for skin perfusion dynamics during wound healing. Potential underlying mechanisms include the metabolic and neurogenic activity associated with the metabolic regulation of blood flow [9,10].

## 3. Discussion

Wound healing is a complex multistage process involving many molecular interactions and signaling pathways that permit successful repair of the skin. Thy-1 has been considered a protein of interest in this process because its upregulation during the inflammatory stage has been associated with angiogenesis [16,17]. Here, we demonstrated that the lack of Thy-1 in mice retards the wound healing process and impairs re-epithelialization associated with decreased blood perfusion during the healing process. Relevant in this context is the participation of metabolic, myogenic, and neurogenic components of blood perfusion. Indeed, wavelet analysis suggested that metabolic and neurogenic components are the major contributors to the impaired blood perfusion observed during the healing process in *Thy-1* knockout (*Thy-1*^−/−^) mice. Moreover, we observed increased microvessel-to-macrovessel ratios in damaged tissues. In addition, we observed altered skin morphology with disorganized epidermal layers and even the absence of granular and stratum corneum layers in the wounded area. Altogether, these results highlight the relevance of Thy-1 in regulating blood vessel formation and blood flow, revealing this protein as a new wound therapy target.

Thy-1 levels are low in healthy skin; however, levels can increase more than 20-fold, 1–3 days upon injury in endothelial cells [21]. As reported, the *Thy-1* promoter is active at this stage and remains active for up to three weeks, until it starts declining [22]. Upregulated Thy-1 promotes the migration of inflammatory cells; for example, neutrophils—the first cells to arrive in the wounded area—migrate following a chemokine gradient generated by activated platelets, but can also undergo transendothelial migration promoted by the αMβ2 integrin-Thy-1 interaction [16]. A subpopulation of fibroblasts also displays an elevated expression of Thy-1 in injured skin, which potentially differentiates into myofibroblasts inducing tissue contraction [23]. Therefore, it is not surprising that organisms lacking Thy-1 exhibit an altered wound-healing process.

Even though the finding that decreased Thy-1 levels delay the wound healing process has been previously reported in mice [18], those studies used an in vivo model in which Thy-1 expression was blocked with a siRNA in the wounded area. In the present study, similar outcomes were described in an organism that completely lacks this molecule, corroborating that Thy-1 presence is required in the wound healing process. Additionally, our results demonstrated that on day 14, the wound closes similarly in WT and *Thy-1*^−/−^ mice, suggesting that Thy-1 is dispensable for the process in the long run. However, delayed wound healing in mice lacking Thy-1, might have severe consequences for the individual since infections occur during the initial stages of the healing process [24]. Moreover, neutrophils help maintain the damaged area free of pathogens, and delayed arrival of these cells due to Thy-1 absence would favor infection of the area, which could then favor a chronic non-healing wound process.

The re-epithelization speed was slower in the *Thy-1*^−/−^ wounded tissue compared with WT. These findings agree with the results reported by Lee et al. 2013 [18], where Thy-1 siRNA-treated wounds exhibited abnormally delayed re-epithelialization and an altered epidermal structure in the wound area. Like our findings, abnormal re-epithelization was present seven days after injury. Here, we observed the absence of stratum granulosum and stratum corneum Figure 2C(c,d) in the *Thy-1*^−/−^ mouse group. In wounded *Thy-1*^−/−^ mice, the inflammatory phase dependent on blood perfusion was altered. This alteration might explain deficient re-epithelialization since this process depends on tissue perfusion and oxygenation [25]. On the other hand, in a recent report, Shemesh, Fuchs, and co-workers noticed a difference in the epidermis of mice where Thy-1+ stem cells had been ablated [26]. In Figure 2B, unwounded tissue of both mouse groups indicates no major differences in the thickness of the epithelium; therefore, these Thy-1+ stem cells described as cells with a non-redundant function in the epidermis are likely not essential for normal epithelialization. Since the epidermal compartment of the injured skin differs between WT and *Thy-1*^−/−^ mice (Figure 2C), it is possible that the delayed wound closure detected in *Thy-1*^−/−^ mice could involve defects within the epidermal compartment itself. This possibility would require further experimentation.

A critical role in vascular maturation has been ascribed to pericytes. Pericytes subjected to endothelial cell-derived factors, such as PDGF (DD and BB), endothelin-1, TGF-β, and HB-EGF, assemble around endothelial cells for tube formation [27]. Moreover, the association of pericytes and collagen type IV promotes the maturation of microvessels [28], and the promotion of endothelial cell junction and ECM deposition to the vascular basement membrane are vital to maintaining vascular stability and homeostasis [29].

Thy-1 presence or absence in pericytes constitutes an interesting avenue of research. Park and co-workers reported that the lack of Thy-1 in brain pericytes increases the ECM protein deposition in the basal membrane, compared to those not lacking Thy-1. However, the stimulation of Thy-1-positive pericytes with TGF-β1 greatly enhances fibrotic activity, suggesting that perivascular Thy-1-positive pericytes could be involved in fibrotic scar formation [30]. In injured skin, Thy-1-positive fibroblasts are relevant due to their role in tissue contraction [23]. However, in the present study, we provide new evidence regarding impaired blood perfusion, which may implicate alterations in endothelial cells and angiogenesis. These alterations include high basal perfusion in the *Thy-1*^−/−^ mouse group, associated with reduced perfusion (compared to WT mice) during the healing process, with an increased number of blood vessels (i.e*.,* microcirculation) at day seven after injury. Additionally, a significant drop in blood perfusion in the wounded area was observed for both groups of mice, but the decline was even higher in *Thy-1*^−/−^ than in WT mice at day 14 (Figure 5A). Although we did not analyze the underlying mechanisms, the results are indicative of increased vascular remodeling in *Thy-1*^−/−^ mice. Therefore, angiogenesis and vascular remodeling processes may be altered in the *Thy-1*^−/−^ group. Thy-1 could also have a role in microvessel maturation and ECM deposition to form the granulation tissue, a possibility that would explain the morphological differences detected between WT and *Thy-1*^−/−^ mouse skin wounds.

In agreement with this last possibility, we also found changes in the dynamics with which different components of the blood perfusion participated, either basally or during the healing process. A possible interpretation of these results is that although basal levels of blood perfusion are higher in *Thy-1*^−/−^ mice, the lack of Thy-1 decreases the capacity to fully compensate for the required blood perfusion over fourteen days of wound healing. This reduced response in blood perfusion in the *Thy-1*^−/−^ group was associated with more significant changes in the wavelet components in the peripheral area, rather than the wounded area, and was mainly observed in the metabolic and neurogenic components. These changes were to be expected, since injury generates a sensitive (i.e., neurogenic) and reddened (i.e., vasodilation or metabolic) area in the periphery of the wound. Interestingly, the dynamics of the participation of wavelet components suggest that the reduced capacity to fully compensate for the required blood perfusion on day seven observed in the *Thy-1*^−/−^ group is most likely associated with an enhanced metabolic and neurogenic response in both the peripheral and wounded area. Furthermore, participation of the neurogenic component appears more persistent since it remains enhanced up to day fourteen in the peripheral area. This analysis provides the groundwork for future research evaluating the formation of new vessels in *Thy-1*^−/−^ mice.

Thy-1 presence or absence in angiogenesis has been reported by Wen et al., who described that Thy-1 could promote healing in an early stage of wound closure but delay the process if overexpressed or absent [31]. These data correlate with our findings, where after seven days (early stage of wound healing), lack of Thy-1 impaired the process, while at a later stage (fourteen days), the differences between groups were not significant. Despite that, our study provides further insight into the relevance of Thy-1 in regulating wound healing and blood perfusion dynamics, as well as the angiogenic process. However, further research is required to elucidate the molecular mechanisms explaining the role of Thy-1 in these processes.

An exciting and novel finding of the present study was that blood perfusion decreased in the wound area of *Thy-1*^−/−^ mice. This finding could explain the delayed wound healing process and highlights the relevance of Thy-1 in the promotion of wound closure. The lower levels of blood perfusion in wounds observed in *Thy-1*^−/−^ mice, compared with WT mice, could be related to a thicker dermis, where the angiogenesis process occurs, together with a higher microvessel to macrovessel ratio detected in mice lacking Thy-1. In this context, chronic wounds, such as those found in patients with type 2 diabetes or obese individuals [32,33], could perhaps be stimulated to heal faster by treating with soluble Thy-1. However, this is a possibility that is currently being investigated.

During the preparation of this manuscript, a paper stating that Thy-1 absence promotes, rather than delays wound healing, was published by Sedov et al. in Nat Cell Biol [34]. These authors reported that the wound healing process and hair follicle regeneration were accelerated in mice lacking Thy-1 (*Thy-1*^−/−^) due to proliferation dependent on YAP signaling. Of note, this is the first article reporting that Thy-1 deficiency promotes wound healing, while other authors have described delayed wound closure in mice with low levels of Thy-1 in the injury site [18]. Importantly, we corroborate Lee’s findings using *Thy-1*^−/−^ mice, which is the same model used in Sedov’s paper. These opposing results could be explained by the differences in the inflicted wound area (1 cm^2^ versus 2 mm diameter used in the present study, or 4–6 mm diameter used in Lee’s paper [18] and the localization of the wound (in the dorsum versus the region between the ears). The latter is relevant because mice have a thin subcutaneous muscle layer that makes their skin heal mainly by the initial contraction of the wound area [35]. The contraction would be different if the lesion were more extensive and if the location were in a place where the skin is looser (medium dorsum) or tighter (between the ears). Therefore, cells are affected by different mechanical forces, which are known to affect the responses of the Thy-1/integrin/syndecan-4 complex [17]. Unfortunately, the paper by Sedov et al. did not discuss the differences between their findings and those previously reported by Lee and co-workers [18], which demonstrated the opposite effect caused by Thy-1 deficiency. These last results were confirmed in the present report.

## 4. Materials and Methods

### 4.1. Animals

Mice C57BL/6 WT (12) and *Thy-1*^−/−^ (12), male, 6-8 months old, weighing 20–25 g were used. The original mouse colony was kindly donated by Dr. James Hagood from the University of North Carolina at Chapel Hill, NC, USA (received from Kevin Kelley, Mt. Sinai School of Medicine) and kept in the Faculty of Medicine’s facilities according to the bioethics committee protocol, CBA1200-22540-MED-UCH.

In vivo experiments were performed under anesthesia using isoflurane (USP, Baxter, Deerfield, IL, USA) as the inhalation anesthetic at 3% in a mixture with oxygen**.** In addition, 2 mg/Kg ketoprofen (Rhodia Merieux, Paulinia, Brazil) was applied for pain management after injury.

### 4.2. In Vivo Wound Healing Assay

The in vivo wound-healing assay was performed as previously described by our laboratory [36,37], with slight modifications. Briefly, WT and *Thy-1*^−/−^ male mice were maintained separately in individual cages. Animals were anesthetized using 3% Isoflurane mixed with oxygen and maintained with 1.5% Isoflurane during the procedure. Then, the dorsal portion of the head was shaved to generate a full-thickness excisional wound between the ears, using a 2 mm biopsy punch. The depth of the wound was approximately 0.5 mm. Wound closure was measured with a vernier caliper and photographically recorded every day placing a metric label in each photo to normalize the measurement. The measurements were performed until day fourteen, at a controlled temperature (22 °C). The percentage of wound closure was calculated as follows: wound healing = (A0 − An/A0), where wound healing represented wound closure. A0 represented the wound area at time 0, and An, the wound area after “*n*” days of follow-up. We defined the wound area considering the boundary edges of the wound. We also defined a peripheral area, as the zone with erythema surrounding the wound, which was about 5 mm beyond the edges of the wound.

### 4.3. Speckle Laser Perfusion Analysis

Tissue perfusion analysis was performed using the Pericam^®^ PSI-HR system (Perimed Ltd., Stockholm, Sweden), as previously reported by our group [38,39], with some minor changes. The instrument uses an invisible near-infra-red laser to measure blood perfusion. A diffuser spreads the laser beam over the region of interest producing a speckle pattern, which is monitored by a CCD camera. Blood perfusion is calculated by analyzing the variations in the speckle pattern, which generates a color map with a chromatic scale ranging from blue (lowest blood flow) to red (highest blood flow). The analysis matrix included an area of 64 × 64 points. Blood flow was recorded in the dorsal portion of the anesthetized animal’s head (Isoflurane at 3% for 5 min). Both WT and *Thy-1*^−/−^ mice were analyzed side-by-side. Briefly, mice had their dorsal cranial area shaved one day before initiating the tissue perfusion analysis. Such prior depilation avoids the detection of changes in blood perfusion due to skin irritation. In addition, blood perfusion was measured without any pharmacological or physical stimulation. Blood flow was recorded for 5 min in the wound area and normalized to the perfusion in an area located approximately 5 mm away from the wound in the same animal (regions of interest, ROIs). Two different blinded observers analyzed the images to consider inter-observer variability.

### 4.4. Wavelet Analysis

Skin perfusion data were used for the wavelet spectral analysis. Each experimental condition was analyzed every 2 min. By applying the Discrete Fourier Transformation (DFT) function in Excel (Microsoft, Washington, DC, USA), we transformed the data from the time domain to the frequency domain, calculating the amplitude-frequency spectrum of the waveform. This analysis generates a signal intensity distribution profile over a range of frequencies. The wavelet data was sampled every 0.2 s to yield a frequency spectrum between 0.0095–2.5 Hz. This range allowed us to analyze the contribution of each regulatory component in the perfused zone of interest. Each component has a specific band frequency range: metabolic (0.0095–0.016 Hz), neurogenic (0.02–0.06 Hz), myogenic (0.0–0.15 Hz), respiratory (0.15–0.4 Hz), and cardiac bands (0.6–2 Hz). Due to the fact that we sought to identify differences in microcirculation, we analyzed only metabolic, neurogenic, and myogenic wavelets. Once the new frequency spectrums were obtained, each spectrum band contribution was calculated by evaluating the area under the curve. Each contribution was displayed as a percentage of the relative energy of all the analyzed bands.

### 4.5. Histology

Skin samples were fixed in formalin (in PBS, 4%, *v*/*v*) for 48 h. Tissue inclusion in paraffin was performed automatically (Leica Biosystem, Wetzlar, Germany) using a previously described protocol [40]. Subsequently, tissues were sectioned (4 μm) and stained with hematoxylin/eosin. Photos were taken at different magnifications using a Nanozoomer XR tissue scanner, Hamamatsu (REDECA, Universidad de Chile). The histology of samples was evaluated in a blinded manner by a trained veterinarian (J.López.) and an odontologist (MR), considering previous publications [41].

Six individual and random images of tissues stained with hematoxylin/eosin were used to manually estimate the thickness of the dermis and epidermis (in micrometers, µm) from high-resolution photographs taken with the tissue scanner. We used three animals per group on seven or fourteen days after injury. The measurements were performed three times in each image and the epidermis was measured from the margin of the skin to the origin of the dermal layer, whereas the distance from the epidermal ridge to the dermal-fat junction was considered the dermis. The number of blood vessels were also measured from six random images per condition, and three animals per group. In addition, we identified blood vessels in each photo by their morphology and the presence of a lumen. Furthermore, the lumen area of the blood vessels was used to categorize them into macro- or micro-vasculature considering a cut-off area of 100 μm^2^. Values higher than 100 μm^2^ are considered to reflect a macrovessel, and lower than 100 μm^2^, a microvessel [42]. Therefore, values are expressed as the percentage or area of macro and microcirculation on day seven or fourteen after skin injury. Photos were processed using Pro-Plus software (Media Cybernetics, Silver Spring, MD, USA).

### 4.6. Statistical Analysis

Values are presented as the mean ± standard error of the mean (SEM) and percentage, where appropriate, and were compared between groups using Mann-Whitney’s U-test. For perfusion analysis, values are presented as the mean and SEM. Differences between WT and *Thy-1*^−/−^ mice were also compared using non-parametric analysis. Data and statistical analyses were performed using GraphPad Prism 6.00 (GraphPad Software, San Diego, CA, USA).

## Figures and Tables

**Figure 1 ijms-23-12539-f001:**
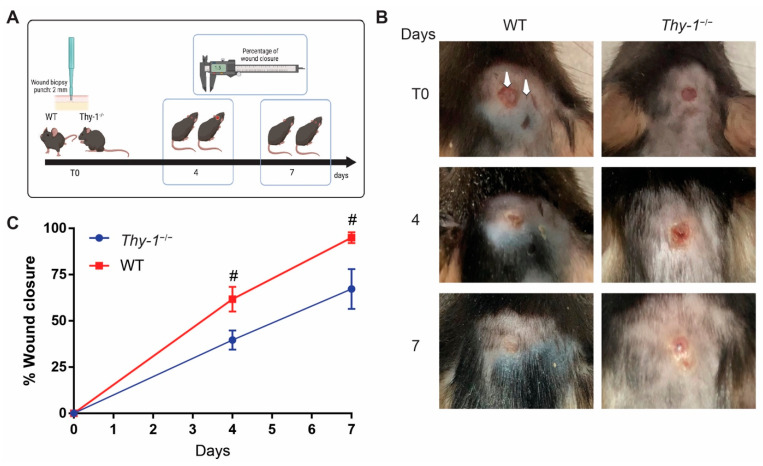
A lack of Thy-1 expression delays wound healing. (**A**) Workflow: a wound biopsy punch of 2 mm was applied on day 0 (T0), and the wound area was measured on days four and seven. (**B**) Representative images from WT and *Thy-1*^−/−^ mice, challenged with the 2 mm biopsy punch at T0 and after four and seven days. The left and right arrows indicate the wounded area and normal skin, respectively. (**C**) Percentage of wound closure over time. Values in the graph correspond to the mean ± SEM (*n* = 4). # *p*-value < 0.05, compared with *Thy-1*^−/−^ at the same time point.

**Figure 2 ijms-23-12539-f002:**
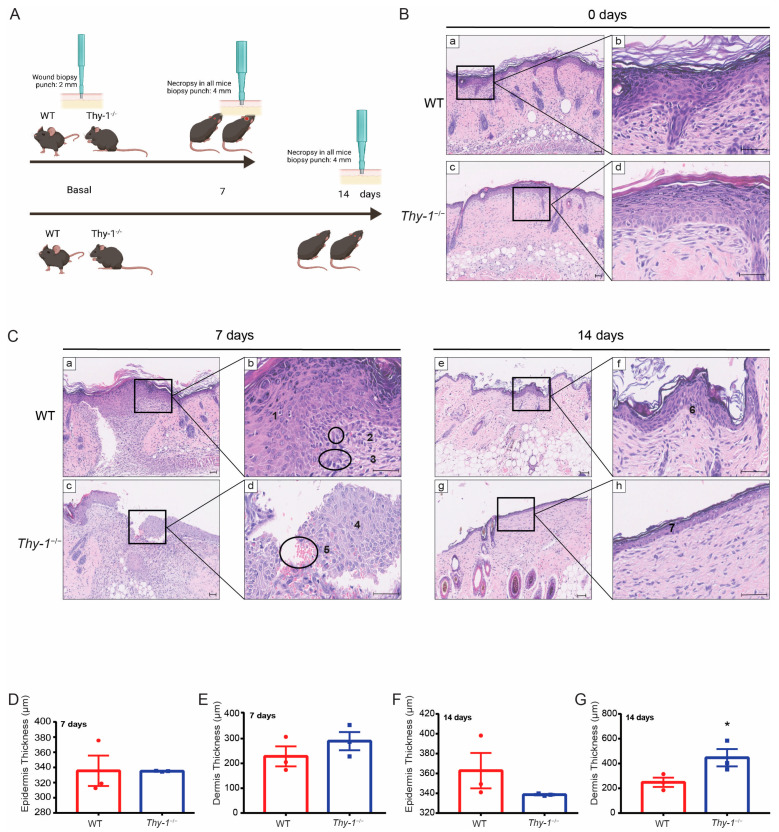
Histological skin analysis in the wounded area of WT and *Thy-1*^−/−^ mice. (**A**) Workflow: after injury with a 2 mm biopsy punch, tissue was collected after seven and fourteen days with a bigger biopsy punch (4 mm). (**B**) Representative images of the histological analysis on the day before the skin injury in WT and *Thy-1*^−/−^ mice at 10× (**a**,**c**) and 40× (**b**,**d**) magnification. (**C**) Representative images of the histological analysis on days seven and fourteen after skin injury in WT and Thy-1^−/−^ mice at 10× (**a**,**c**,**e**,**g**) and 40× (**b**,**d**,**f**,**h**) magnification. (**a**,**b**) Complete re-epithelization on day seven in WT mice; all epidermis layers are present. (1 = stratified squamous epithelial, 2 = microvessels, 3 = fibroblast-like cells). (**c**,**d**) Incomplete re-epithelization on day seven in *Thy-1*^−/−^ mice (4 = no stratum granulosum and no stratum corneum are present, 5 = extravascular red blood cells); (**e**–**h**) Stratified squamous epithelium is restored to its original size by day fourteen (6 and 7 = complete re-epithelization); no major histological differences were observed between the two groups. Scale bar = 50 µm. (**D**–**G**) Estimation of the epidermis (**D**,**F**) and dermis (**E**,**G**) thickness in the wounded area on day seven or day fourteen after injury in WT or *Thy-1*^−/−^ mice, respectively. * *p*-value < 0.05.

**Figure 3 ijms-23-12539-f003:**
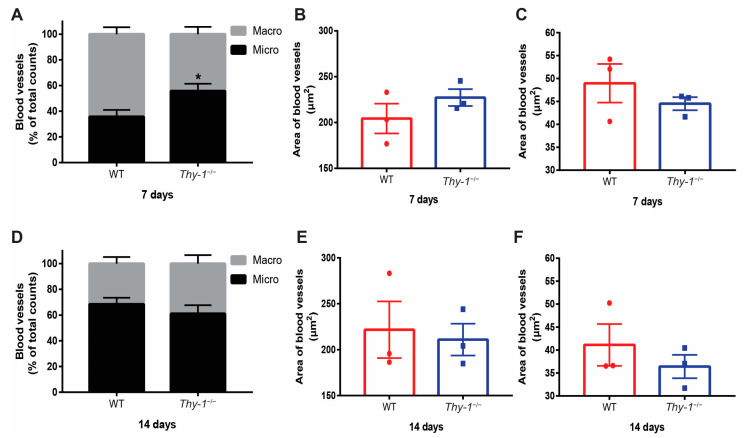
Estimating macro and microvascular blood vessels in the wounded area of WT and *Thy-1*^−/−^ mice. (**A**) Estimation of the percentage of macro and microvascular blood vessels in the wounded area on day seven after injury in WT (*n* = 3) and *Thy-1*^−/−^ mice (*n* = 3). (**B**) Estimation of the area of macrovascular; or (**C**) microvascular blood vessels on day seven after injury in WT and *Thy-1*^−/−^ mice. (**D**) Estimation of the percentage of macro and microvascular vessels (using a cut-off of 100 μm^2^) as in A, but on day fourteen after injury (*n* = 3, per group). (**E**,**F**) Estimation of the area of macrovascular (**E**) or microvascular (**F**) blood vessels on day fourteen after injury in WT and *Thy-1*^−/−^ mice. Values are presented as mean ± SEM * *p*-value < 0.05.

**Figure 4 ijms-23-12539-f004:**
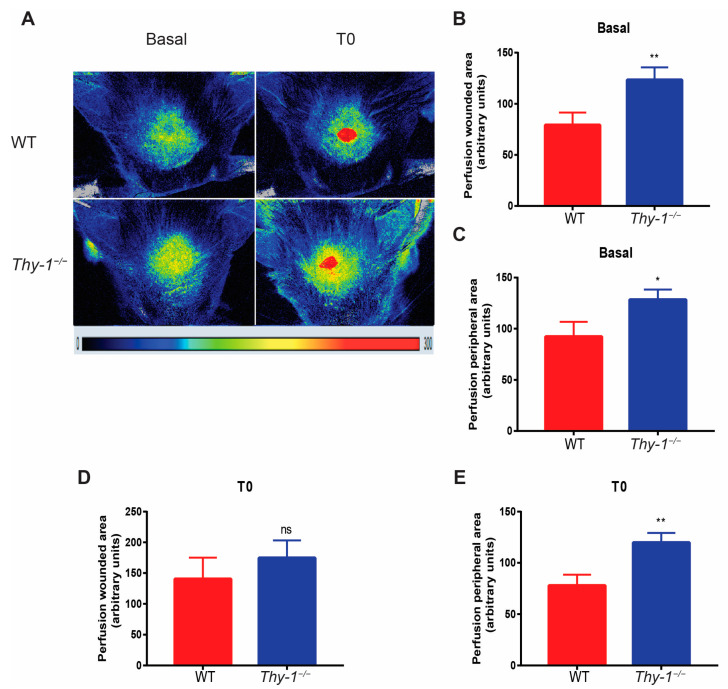
Blood perfusion in the wounded area in WT and *Thy-1*^−/−^ mice. (**A**) Representative images of blood perfusion in the wounded area in WT (*n* = 10, upper) or *Thy-1*^−/−^ mice (*n* = 8, bottom). Representative images of basal blood perfusion before skin injury, where T0 represents blood perfusion immediately after skin injury. Quantification of basal blood perfusion in the wounded area (**B**) and peripheral area (**C**) is shown. For T0, the quantification of blood perfusion in the wounded area (**D**) and peripheral area (**E**) is presented. Values are presented as mean ± SEM. ns, non-significant; * *p*-value < 0.05, ** *p*-value < 0.01 versus WT mice.

**Figure 5 ijms-23-12539-f005:**
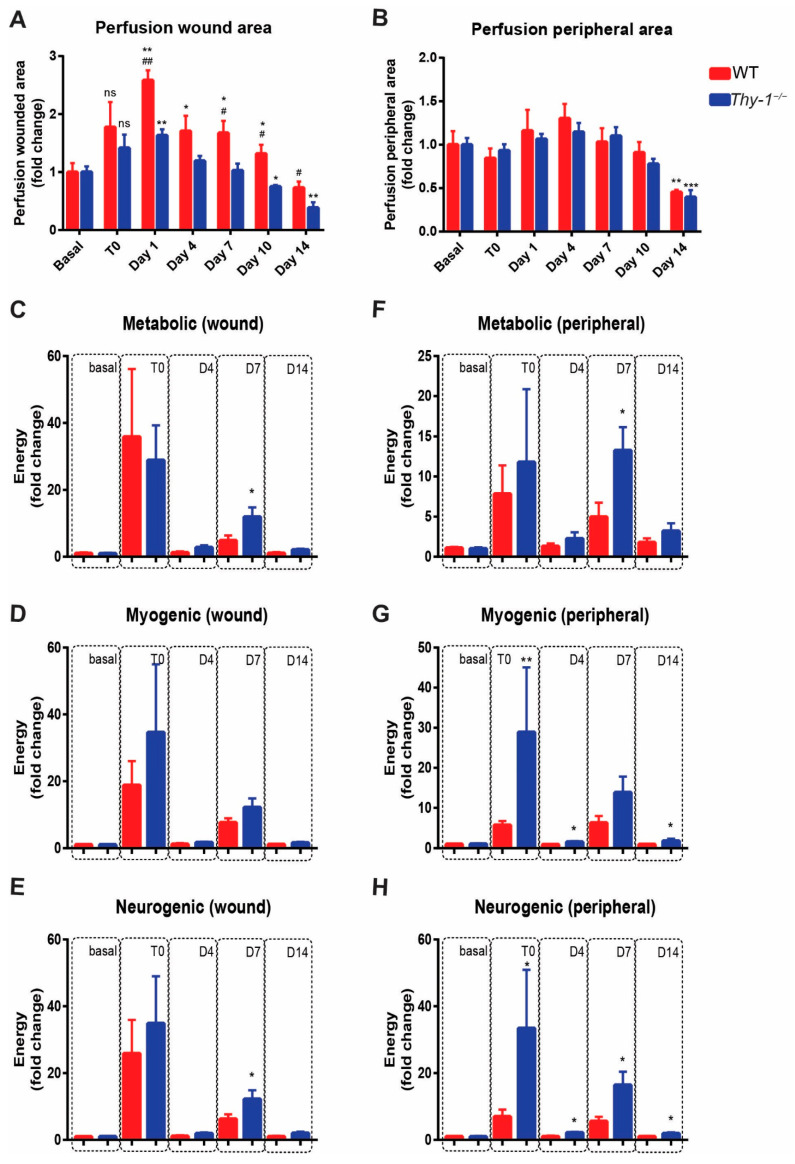
Blood perfusion analysis in the wounded and peripheral areas. Blood flow was measured under basal conditions, immediately after injury (T0), and 1, 4, 7, 10, and 14 days after introducing the biopsy punch, in (**A**) wounded area and (**B**) peripheral area, both in WT and *Thy-1*^−/−^ mice. (**C**–**E**) Energy signal filtered and analyzed separately for the Metabolic, Myogenic, and Neurogenic components in the wounded area. (**F**–**H**) Energy signal filtered and analyzed for the Metabolic, Myogenic, and Neurogenic components in the peripheral area. Values are presented as mean ± SEM * *p*-value < 0.05; ** *p*-value < 0.01 and *** *p*-value < 0.001. All of them compare the WT and *Thy-1*^−/−^ groups at each timepoint; # *p*-value < 0.05; ## *p*-value < 0.01 and both compare values for the same condition at different time points with basal levels; ns = no significance.

## Data Availability

Any additional information required to reanalyze the data reported in this paper is available from the authors upon request.

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
