# Peer review of "The GPI-Anchored Protein Thy-1/CD90 Promotes Wound Healing upon Injury to the Skin by Enhancing Skin Perfusion"

_ijms, 2022, doi:10.3390/ijms232012539_

Round 1
Reviewer 1 Report
In this manuscript, the authors demonstrate a role of THY-1/CD90 in the regulation of the wound healing process, through a mechanism that involves angiogenesis and blood perfusion dynamics. My opinion is favorable for publication. However, there are some concerns that should be adressed.
1. I think that the main weakness of this study is that the consequences of the lack of Thy-1 in the wound healing process are exclusively interpreted and discussed from the aspect of the angiogenesis process, which is conceivable since this is the focus of the study. However, Thy-1 is a cell-surface antigen that is expressed in other cell types of the skin, such as fibroblasts that play an important role in the wound healing process. It is quite expected that in a knockout Thy-1 animal model all expressing cells in a tissue may be affected and therefore may affect the wound healing process. I think that this information should be included in the introduction part.
2. Considering the experimental section:
- the method used to manually quantify epidermal and dermal thickness could be better explained.
-Figure 4. Blood perfusion is proportional to the red color intensity. Could you give some more details about the method of quantification?
3. In Figure 2 there are histological sections of the skin of both mice groups after wound closure. It is interesting that in the Thy-/- type the epidermis is almost totally flat (no rete ridges contrary to the WT) and the distinct epidermal layers are not clearly visible as commented also by the authors. This result implies that there is a defect in the epidermal differentiation process in the Thy-1-/- model after re-epithelialization and likely an accelerated maturation process. Could this be explained solely by a defect in blood perfusion?This raises the question: do the authors see a difference in the quality of epidermis of unwounded WT and Thy-1 -/- mice? Maybe it wound be interesting to add an histological section of undamaged skin of both groups. Furthermore, I think that the delay in the re-epithelialization process might also be related with defects within the epidermal compartment itself. This should be further discussed as well as other possibilities.
For example, there is a recent article of Koren et al showing that Thy-1 is expressed in a subpopulation of low-cycling basal epidermal progenitors with a significant expansion potential in vitro that contribute to wound repair.
Koren E, Feldman A, Yusupova M et al Thy-1 marks a distinct population of slow-cycling stem cells in the mouse epidermis. Nat Commun 13, 4628 (2022).
4. Do the authors have any idea why the Thy-/- mice present a higher basal blood perfusion than the WT?
5. I think that the discussion section should be reformatted or enhanced according to the previous comments.
Author Response
REVIEWER #1
Open Review
In this manuscript, the authors demonstrate a role of THY-1/CD90 in the regulation of the wound healing process, through a mechanism that involves angiogenesis and blood perfusion dynamics. My opinion is favorable for publication. However, there are some concerns that should be adressed.
We thank this reviewer for the favorable opinion of our manuscript.
- I think that the main weakness of this study is that the consequences of the lack of Thy-1 in the wound healing process are exclusively interpreted and discussed from the aspect of the angiogenesis process, which is conceivable since this is the focus of the study. However, Thy-1 is a cell-surface antigen that is expressed in other cell types of the skin, such as fibroblasts that play an important role in the wound healing process. It is quite expected that in a knockout Thy-1 animal model all expressing cells in a tissue may be affected and therefore may affect the wound healing process. I think that this information should be included in the introduction part.
This reviewer is correct when noticing that other cells, which also increase Thy-1 levels upon injury, such as fibroblasts, have been omitted. Thus, we have introduced in the Discussion section, information regarding the role of fibroblasts in the process of wound healing. Besides, we also mentioned that our focus of interest is on endothelial cells and angiogenesis due to our findings relating to skin perfusion.
Sentences now read:
In the second paragraph of the Discussion section (line 187): “A subpopulation of fibroblasts also displays elevated expression of Thy-1 in injured skin, which potentially differentiate into myofibroblasts inducing tissue contraction [23].”
Then in the sixth paragraph of the Discussion section (line 221): “In injured skin, Thy-1-positive fibroblasts are relevant due to their role in tissue contraction [23]. However, in the present study, we provide new evidence regarding impaired blood perfusion, which may implicate alterations in endothelial cells and angiogenesis. These alterations include…”
- Considering the experimental section:
- the method used to manually quantify epidermal and dermal thickness could be better explained.
We have clarified this method in the section of Materials and Methods subtitled Histology and included the following, “Six individual and random images of tissues stained with hematoxylin/eosin were used to manually estimate the thickness of the dermis and epidermis (in micrometers, µm) from high-resolution photographs taken with the tissue scanner. We used 3 animals per group at day 7 or 14 after injury. The measurements were performed three times in each image and the epidermis was measured from the margin of the skin to the origin of the dermal layer, whereas the distance from the epidermal ridge to the dermal-fat junction was considered the dermis. The number of blood vessels were also measured from six random images per each condition, and 3 animals per group.” Line 330.
-Figure 4. Blood perfusion is proportional to the red color intensity. Could you give some more details about the method of quantification?
The use of the speckle laser Doppler was performed using the Pericam® PSI-HR system (Perimed Ltd., Stockholm, Sweden), as indicated in M&M and as explained in our previous publications [33,39]. This device uses an invisible near-infra-red (NIR) laser for blood perfusion measurements. The light source employed is a solid-state laser with a wavelength of 785 nm, power of 80 mW maximum output power, and polarized with a polarizer in front of the camera. A diffuser spreads the beam over the measurement area, creating a speckled pattern, which is monitored in the illuminated area using a 2448 x 2048-pixel CCD camera (charge coupled device) that can take images at a speed of up to 120 frames per second. Blood perfusion is calculated by analyzing the variations in the speckle pattern that generate a color map with a chromatic scale ranging from blue (reduced blood flow) to red (high blood flow). For both wild type and Thy-1 -/- mice, measurements were obtained side-by-side.
The Methods section subtitled Speckle laser perfusion analysis has been modified accordingly and newer references from our group (38,39) have been incorporated to this section.
- In Figure 2 there are histological sections of the skin of both mice groups after wound closure. It is interesting that in the Thy-/- type the epidermis is almost totally flat (no rete ridges contrary to the WT) and the distinct epidermal layers are not clearly visible as commented also by the authors. This result implies that there is a defect in the epidermal differentiation process in the Thy-1-/- model after re-epithelialization and likely an accelerated maturation process. Could this be explained solely by a defect in blood perfusion?This raises the question: do the authors see a difference in the quality of epidermis of unwounded WT and Thy-1 -/- mice? Maybe it wound be interesting to add an histological section of undamaged skin of both groups. Furthermore, I think that the delay in the re-epithelialization process might also be related with defects within the epidermal compartment itself. This should be further discussed as well as other possibilities.
We thank the reviewer for these valuable comments; new images with hematoxylin-eosin stained tissue have been added to figure 2, which represents healthy skin from both mouse groups (now Figure 2B). The corresponding text has been added to the main body of the manuscript and to the figure legend, lines109 and 711, respectively. From these newly added images in Figure 2B, it is clear that there are no major differences between the skin tissue of both animal groups before the wounding.
In the wounded Thy-1-/- mouse group, the inflammatory phase, which depends on blood perfusion, is altered. The alterations result in deficient re-epithelialization since this process depends on tissue perfusion and oxygenation. However, since the epidermal compartment of the injured skin differs between WT and Thy-1-/- mice, the delayed wound closure detected in Thy-1-/- mice might be also due to defects within the epidermal compartment itself, as suggested by this reviewer. Assessing this possibility would require further experimentation that goes beyond the scope of the present study. A similar statement has been added to the Discussion section. Line 207.
For example, there is a recent article of Koren et al showing that Thy-1 is expressed in a subpopulation of low-cycling basal epidermal progenitors with a significant expansion potential in vitro that contribute to wound repair.
Koren E, Feldman A, Yusupova M et al Thy-1 marks a distinct population of slow-cycling stem cells in the mouse epidermis. Nat Commun 13, 4628 (2022).
We thank this reviewer for pointing out this recently published paper to us. It is very interesting that these authors noticed a difference in the epidermis of the mice where Thy-1+ stem cells had been ablated. We have now added a new figure to include unwounded tissue of both mouse groups (Figure 2B) where no major differences are observed in the thickness of the epithelium; therefore, these Thy-1+ stem cells with a non-redundant function are likely not essential for normal epithelialization. However, the injured tissue displays clear differences between the WT and the Thy-1-/- skin, as described in Figure 2.
As pointed out by this reviewer, the delay in the re-epithelialization process might also be explained by defects within the epidermal compartment itself, but the association with defects in blood perfusion, as we describe in the present manuscript, are also important since blood-borne factors, as well as oxygenation, are required for the re-epithelialization process. We have added a sentence to the discussion to include these alternative interpretations to the data. Line 207.
- Do the authors have any idea why the Thy-/- mice present a higher basal blood perfusion than the WT?
This result is intriguing, because no information is available regarding the potential role of Thy-1 in the regulation of the vascular tone. Moreover, Thy-1 is present on endothelial cells and positively regulates angiogenesis (i.e., new blood vessel formation) only under inflammatory conditions, namely in “activated” endothelial cells. Further studies would be required to better understand this finding.
- I think that the discussion section should be reformatted or enhanced according to the previous comments.
As requested, numerous changes have been incorporated into the Discussion section.

Reviewer 2 Report
The study is interesting because, unusually, it combines biochemical measures, biophysical measures and
anatomical outcome measures. As such it addresses a commonly overlooked gap between lab-bench measures
and applied clinical measures.
The study is better than average; however, it does include some errors, subjectivity, and ambiguous language. I
find several instances where data is overinterpreted. In my opinion, several alternative explanations exist that
could also explain the data; however, are not considered, nor discussed. Despite these weaknesses, I believe the
work will add value to the literature; it should be published, but only after the errors and ambiguities are addressed.
Ln 33. Why is an article published in a specialist Dental journal cited to substantiate “skin wounds… chronic ulcers,
inflammation, and persistent infections…”? Healing in oral and buccal mucosa is not equivalent to healing in
cutaneous skin! Please cite more appropriate data.
Ln 35. Please justify citing “Gnyawali et al 2015, 2017”, as substantiating evidence for “Wound healing involves
multiple processes…”? I am not convinced that articles reporting the application of advanced imaging technologies
is acceptable. Have you considered Singer and Clark (1999. doi: 10.1056/NEJM199909023411006); or Price,
Grey, Patel, Harding (2021. ISBN: 978-0-470-65897-0)?
Ln 39. What is the evidence for “impaired vascular function” claimed to be reported in “Gnyawali et al 2015, 2017”?
I find no such substantiating data. These are not appropriate citations.
Ln 96. Figure 1. It is not clear what type of wound was created: “…a wound biopsy punch of 2 mm was applied…” I
might interpret this to be an excisional wound; however, what depth? Is this wound superficial (i.e. epidermis)?
partial thickness (i.e. dermis)? or, is this wound full thickness (i.e. hypodermis)? How might others reproduce this
wound?
Ln 105. What is meant by “virtually closed”? The adjective “virtually” has several meanings in modern usage.
Ln 106. What is the meaning of “looked different”? This is not an objective measure.
Ln 107. I have the same issue with “tissue of the wound area in WT mice was also different from that of Thy-1-/-
mice”. What is the meaning of “different”? Please remove subjective measures!
Ln 129. What is the evidence “WT mice healed faster than Thy-1-/- mice”?
Ln 140. There are no scale bar or indicators of size evident. It is difficult to verify histological data without some
indication of size.
Ln 140. What is the measure “Epidermis Length”; “Dermis Length”? Length is a lateral parameter. Does the author
mean epidermis thickness/depth; dermis thickness/depth?
Ln 144. What is the evidence “2= angiogenesis, 3= fibroblasts”? Morphology is not sufficient to identify the origin of
individual cells.
Ln 146. What is the evidence “Stratified squamous epithelium is restored to its original size by day 14”? The
indicated fields are not equivalent sizes. Importantly, I find the dermal organisation is not equivalent. Not only are
these images from different (not equivalent) locations, but they are also not reported with equivalent magnification.
This is data misrepresentation!
Ln 146. What is “extravasated blood vessels”? Does the author mean 'ruptured blood vessels’? …or, perhaps they
mean ‘extravascular erythrocytes (blood)”?
Ln 152. It not clear how the authors “differentiate macro and microvessels”? What is the criteria for a macro
vessel? What is the criteria for a micro vessel?
Ln 166. It is not clear how many vessels were counted? Is “n=3” refer to: the number of wounds? …the number of
biopsies? …the number of animals? …the number of experiments?
Ln 169. What is the quantitative parameter “area of macrovascular…or microvascular… blood vessels”? Does this
refer to lumen area? …total area of endothelial cells? …area occupied by lumen structures?
Ln 174. The function of the blood clot is not “to fill the wounded area”! The function of thrombogenesis is to restore
haemostasis!
Ln 175. What is the evidence “Clots also aid in eliminating cytotoxic substances”?
Ln 189. The phrase “exhibited a tendency” is subjective. Please replace with objective measures.
Ln 220. It is evident that Thy-1–/– animals exhibit higher basal perfusion than WT animals, irrespective of
intervention. Why are Thy-1–/– animals compared with Thy-1+/+ animals after injury? Is it not be more relevant to
compare pre-wound perfusion with post-wound perfusion in the same animal?
Ln 216. What is the evidence “Thy-1 is required to regulate skin perfusion dynamics during wound healing”? I
accept the evidence indicates that Thy-1 has a role in skin perfusion dynamics, but what is the basis to conclude
Thy-1 contributes regulatory function? …And why only during wound healing?
Ln 242. What is the evidence “Thy-1… controls angiogenesis”?
Ln 243. I take exception with the statement “blood perfusion (is) associated with wound closure during the healing
process.” Wound closure in mammals is defined as complete reepithelialisation and reestablishment of epidermal
barrier function. Granulation and angiogenesis are consequential; they do not directly contribute to wound closure.
This confusion may be a misinterpretation of cutaneous healing in mice and humans; where the wound healing
processes are not equivalent.
Ln 245. What is the evidence “wavelet analysis suggested that metabolic and neurogenic components are the
major contributors to the impaired blood perfusion observed during the healing process in Thy-1 knockout mice”?
Ln 249. What is the data substantiating “altered skin morphology with disorganized epidermal layers and even the
absence of granular and stratum corneum layers in the wounded area”?
Ln 273. What is the data substantiating “re-epithelization speed was slower in the Thy-1-/- wounded tissue”?
Ln 277. The evidence reported to substantiate the claim for “absence of stratum granulosum and stratum corneum
in the Thy-1-/- mice group” is weak. The data at 14 days is unambiguous evidence Thy-1–/– animals develop both
stratum granulosum and stratum corneum. The data at 4 days is not clear due to selective cropping of Thy-1+/+
animal data. In my opinion, these data are overinterpretted. As wound closure is delayed in Thy-1–/– animals, so too squamous differentiation is also delayed.
Ln 331. An alternative explanation to the observation that “lower levels of blood perfusion in wounds observed in
Thy-1-/- mice, compared with WT mice, could be related to a thicker dermis, where the angiogenesis process
occurs, together with a higher microvessel to macrovessel ratio detected in mice lacking Thy-1” is that the wound
bed of Thy-1–/– mice is not capable of restoring a functional extracellular matrix due to dysfunction (read absence)
of interactions between cell surface adhesive receptors, proteoglycans, glycosaminoglycans and glycolipids – of
which Thy-1 is just one of several. Thus the scaffolding required by neo-endothelium, neo-neurothelium, neo-mesenchyme and neo-epithelium is absent, is aberrant, and/or is also dysfunctional. This hypothesis predicts
that Thy-1 contributes a critical functional component within this complex orchestra of tissue specific events and processes.
However, this hypothesis does not invoke the concept that soluble Thy-1 might be applied to stimulate faster
healing in pathological skin tissue (e.g. diabetes), as discussed in the text.
Ln 364. It is stated in Materials and Methods that animals were anaesthetised “using isofluorane… as the inhalation anaesthetic at 3% in a mixture with oxygen." Why does the very first experiment described in Ln 370 (Animals were anesthetized using 2-5% Isoflurane mixed with oxygen) deviate from this protocol?
Ln 372. It is not clear how wound boundaries were determined reliably “…with a vernier caliper and
photographically recorded every day”? Eschar and erythema are evident in the provided images; where is the
wound boundary?
Ln 389. What is the evidence that “perfusion in an area located less than 1 cm away in the same animal” is
equivalent to perfusion in the dorsal cranial area? I am not a mouse anatomist, but it strikes me that in young
mice one centimetre distance from the dorsal cranial area places the Pericam® probe into a distinctly different
anatomical zone.
Ln 417. In the interest of reproducibility, please elaborate how “…the area in which the blood vessels were
counted was measured”?
Ln 420. This reader is confused. It states that macrovessels are “higher than 100 μm2”, and microvessels are “over
than 100 μm2” (sic)? Surely microvessels should be defined to be ‘less than’ 100 μm2?
I also remain confused regarding the measure “area of the blood vessels”? Does this refer to the total area of
tissue populated with blood vessels, or does this refer to the (cumulative?) measured area of individual vessel
lumens?
I remain unconvinced that “area of the blood vessels” is a reliable measure of perfusion capacity of tissue.
Ln 426. Did the authors really apply “Mann-Whitney’s t-test” to “compared between groups”? In this reviewer’s
understanding, the Mann-Whitney U test is used to compare differences between two independent groups when
the dependent variables are NOT normally distributed. In contrast a t-test is used to compare the means of two
groups when the dependent variable IS normally distributed. This statement is self-contradictory.

Round 2
